# Are All Steps Equally Important?
# Benchmarking Essentiality Detection in Event Processes

**Haoyu Wang[1], Hongming Zhang[1], Yueguan Wang[2,3], Yuqian Deng[1],**
**Muhao Chen[3], Dan Roth[1]**

[1]Department of Computer and Information Science, UPenn
[2]Department of Electronic Engineering, THU
[3]Department of Computer Science, USC

{why16gzl,hzhangal,yuqiand,danroth}@seas.upenn.edu,
wangyuel18@mails.tsinghua.edu.cn, muhaoche@usc.edu

## Abstract

Natural language expresses events with varying granularities, where coarse-grained events (goals) can be broken down into finer-grained event sequences (steps). A critical yet overlooked aspect of understanding event processes is recognizing that not all step events hold equal importance toward the completion of a goal. In this paper, we address this gap by examining the extent to which current models comprehend the essentiality of step events in relation to a goal event. Cognitive studies suggest that such capability enables machines to emulate human commonsense reasoning about preconditions and necessary efforts of everyday tasks. We contribute a high-quality corpus of (goal, step) pairs gathered from the community guideline website WikiHow, with steps manually annotated for their essentiality concerning the goal by experts. The high inter-annotator agreement demonstrates that humans possess a consistent understanding of event essentiality. However, after evaluating multiple statistical and large-scale pre-trained language models, we find that existing approaches considerably underperform compared to humans. This observation highlights the need for further exploration into this critical and challenging task[1].

## 1 Introduction

As a fundamental semantic primitive unit in human language ([Jackendoff, 1992](#)), events play a pivotal role in facilitating efficient communication among humans and safe interactions with the world. Recently, the natural language processing (NLP) community has made significant strides in helping machines comprehend events through various directions, such as event extraction ([Grishman et al., 2005](#); [Lin et al., 2020](#)), event relation extraction ([Ning et al., 2018a](#); [Wang et al., 2020a](#)), event schema induction ([Chambers, 2013](#); [Dror](#)

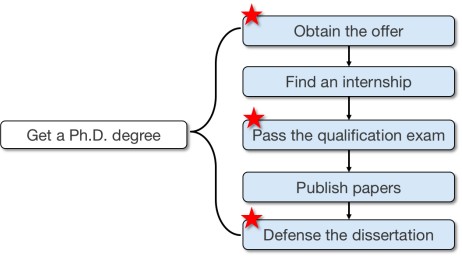

Figure 1: Illustration of steps to obtain a Ph.D. degree, with essential steps marked by red stars. Successfully achieving the overall goal typically necessitates the completion of these crucial steps.

et al., 2023), and event-centric knowledge graph construction ([Tandon et al., 2015](#); [Zhang et al., 2021a](#)). However, most of these studies primarily concentrate on modeling *horizontal* relationships between events, neglecting the internal components of an event (i.e., how an individual perceives an event mention).

Computational and cognitive studies ([Schank and Abelson, 1977](#); [Zacks and Tversky, 2001](#)) indicate that humans can deconstruct a goal event into a discrete representation of finer-grained step events, ultimately facilitating the hierarchical organization of event-related knowledge. As illustrated in Figure 1, when discussing the goal event of *"obtaining a Ph.D. degree"*, we understand that several steps may occur along the way. For instance, one might *receive the offer*, *pass the qualification exam*, *complete internships*, *publish papers*, and *defend the dissertation*. Among these steps, some are deemed essential to the goal, while others are not. For instance, passing the qualification exam is crucial for earning a Ph.D. degree, whereas securing an internship is often not a requirement. This ability to discern the essentiality of steps pertaining to various goals equips humans with the commonsense needed to address problems and carry out daily tasks. Similarly, understanding which steps are essential can profoundly benefit numerous NLP applications. For instance, event

---

[1]The dataset and code are available at http://cogcomp.org/page/publication_view/1023.

schema induction (Dror et al., 2023) relies on event-centric information extraction to derive graphical representations of events from text. In this context, understanding essentiality can enhance the quality of induced schemas by eliminating hallucinations and suggesting the addition of missing crucial events. Moreover, grasping essentiality can potentially benefit intelligent systems for QA tasks (Bisk et al., 2020) and task-oriented dialogue processing (Madotto et al., 2020).

In this paper, we aim to assess the depth of understanding that current NLU models possess regarding events in comparison to human cognition. To accomplish this, we introduce a new cognitively inspired problem of detecting essential step events in goal event processes and establish a novel benchmark, Essential Step Detection (ESD), to promote research in this area. Specifically, we gather goals and their corresponding steps from WikiHow[2] and manually annotate the essentiality of various steps in relation to the goal. Our experimental findings reveal that although humans consistently perceive event essentiality, current models still have a long way to go to match this level of understanding.

## 2 Task and Data

The essential step detection task is defined as follows: for each goal $G$ and one of its sub-steps $S$, the objective is to predict whether the failure of $S$ will result in the failure of $G$. In our formulation, $G$ and $S$ are presented as natural language sentences. The construction of ESD includes two steps: (1) Data Preparation and (2) Essentiality Annotation. Details of these steps are provided below.

### 2.1 Data Preparation

WikiHow is a widely-used and well-structured resource for exploring the relationship between goal-oriented processes and their corresponding steps (Koupaee and Wang, 2018; Zhang et al., 2020b). To the best of our knowledge, it is the most appropriate resource for the purpose of our research. Consequently, we begin by collecting 1,000 random goal-oriented processes from WikiHow. To avoid oversimplified and overly complex processes, we only retain those with three to ten steps. Furthermore, given that all WikiHow processes and their associated steps are carefully crafted by humans, the majority of the steps mentioned are essential.

|  | Essential | Non-essential | Total |
|---|---|---|---|
| Number of instances | 1,118 | 397 | 1,515 |
| Average step length | 17.1 | 17.4 | 17.2 |

Table 1: Dataset statistics of ESD. The average step length represents the mean number of tokens per step.

To achieve balance in the dataset, we enlist crowd-sourcing workers to contribute optional steps (i.e., those that could occur as part of the process but are not essential)[3]. We employ three annotators from Amazon Mechanical Turk[4], who are native English speakers, to provide optional steps for each goal. To ensure high-quality annotations, we require annotators to hold the "Master annotator" title. The average cost and time for supplying annotations are 0.1 USD and 32 seconds per instance (approximately 12 USD per hour).

### 2.2 Essentiality Annotation

Given that our task necessitates a profound understanding of the events and careful consideration, we ensure annotation quality by employing three well-trained research assistants from our department rather than ordinary annotators to conduct the essentiality annotations. For each goal-step pair, annotators are asked to rate it as 0 (non-essential), 1 (essential), or -1 (the step is not a valid step for the target goal, or the goal/step contains confidential or hostile information)[5]. Since all annotators are well-trained and fully comprehend our task, we discard any pair that is deemed invalid (i.e., -1) by at least one annotator. This results in 1,515 pairs being retained. We determine the final label based on majority voting. The dataset statistics can be found in Table 1. Altogether, we compile 1,118 essential and 397 non-essential "goal-step" pairs. The inter-annotator agreement, measured by Fleiss's Kappa[6], is 0.611, signifying the high quality of ESD.

## 3 Experiments

Recently, large-scale pre-trained language models have exhibited impressive language understanding capabilities. To assess the extent to which these models truly understand events, we evaluate them using ESD. Specifically, we benchmark their performance by examining a range of inference meth-

---

[2]WikiHow is a community website featuring extensive collections of step-by-step guidelines.

[3]The survey template is shown in Appendix Figure 2.
[4]https://www.mturk.com/
[5]The survey template is shown in Appendix Figure 3.
[6]We utilize tools from https://github.com/Shamya/FleissKappa.

ods as detailed below.

1. **Next Sentence Prediction**: To ensure that the essentiality detection task aligns with the training objectives of pre-trained masked Language Models (LMs) (Devlin et al., 2019), we verbalize each pair of goal $G$ and step $S$ into two sentences: "To $G$", and "we must $S$". Then we leverage the LM to predict the probability of "we must $S$" to be the next sentence of "To $G$".

2. **Perplexity**: We also attempt to verbalize a goal-step pair into sentences and employ the perplexity predicted by the Language Model (i.e., GPT-2 (Radford et al., 2019)) as an indicator for the predicted perplexity.

3. **Intent Detection**: We assess the performance of an Intent Detection model (Zhang et al., 2020b), which is designed to predict the correct intent given an utterance. By setting $S$ as the provided utterance, we employ the model to predict its corresponding goal $G$.

4. **Textual Entailment**: Another alternative is to leverage the logical inference capabilities of a textual entailment model (Williams et al., 2018). In our experiment, we treat $G$ as the premise and $S$ as the hypothesis. If a model understands the essentiality of completing $S$ in order to achieve $G$, it should be able to infer $S$ from $G$.

5. **Unified QA**: We also experiment with a SOTA QA model. Specifically, We follow the setting in Unified QA (Khashabi et al., 2020) to convert each goal-step pair into a "Yes/No" question and then use the predicted probability of "Yes" as the indicator for the essentiality.

6. **Prompt with GPT-3 & GPT-4**: We test the prompt-based methods as well, which have proven to be powerful for many NLU tasks. Specifically, we manually design templates to convert each goal-step pair into prompts and then ask GPT-3 (Brown et al., 2020) & GPT-4 (OpenAI, 2023) to generate True or False labels based on our input.

7. **Corpus Statistics**: Last, we present the performance of a corpus statistics model to determine whether such knowledge is explicitly expressed in free text. For each goal-step pair, we first extract the central verbs from the goal and step using dependency parsing tools as their representatives. Subsequently, we employ their normalized co-occurrence frequencies in the New

| Model | | Full | Core |
|---|---|---|---|
| Random | ‖ | 0.5000 | 0.5000 |
| Corpus Statistics | | 0.5043 | 0.4987 |
| Next Sentence Prediction | ‖ | 0.5659 | 0.5503 |
| Perplexity | | 0.5934 | 0.5793 |
| Intent Detection | | 0.5461 | 0.5449 |
| Textual Entailment | | 0.5813 | 0.5630 |
| Unified QA | | 0.6012 | 0.6067 |
| Prompt with GPT-3 | | 0.6358 | 0.6043 |
| Prompt with GPT-4 | | **0.6574** | **0.6283** |
| Human | ‖ | 0.8750 | 0.8100 |

Table 2: Results of essentiality detection on ESD. The best results are highlighted in bold font.

York Times Corpus (NYT) (Sandhaus, 2008) to indicate the relationship between them.

Examples of all utilized templates and prompts can be found in Appendix Table 4. Given that we have formulated the task as a binary choice problem, which differs from the output of the Perplexity and TE models, we evaluate all models based on the AUROC score (Hanley and McNeil, 1982; Narkhede, 2018) to enable a fair comparison. For Perplexity, we use the perplexity score as the predicted essentiality (lower scores are better). For TE and Intent Detection models, we use the predicted probability of "Entailment" and the likelihood of being entailed as the predicted essentiality. The statistics-based model uses normalized frequency as the essentiality indication signal. For all other baselines, we use the predicted probability of "Yes" as the predicted essentiality (higher scores are better). All experiments are conducted using the largest available models and the default hyperparameters.

### 3.1 Result Analysis

In WikiHow, each goal (e.g., "Toast Sunflower Seeds") is typically associated with a modifier (e.g., "Microwave Toasting") to provide a more precise definition. In our experiment, we evaluate whether such a modifier impacts the models' comprehension of the goal by employing two settings: (1) Full: we concatenate the goal and the modifier as the goal input; (2) Core: we only use the original goal as the goal input.

We also report human performance as an upper bound for this task. Specifically, we randomly select 200 instances, ask three ordinary annotators to label them, and report the average performance. As annotators provide binary annotations for the essentiality of a goal-step pair instead of a real value

| Model | Encoder | # Parameter | AUROC |
|-------|---------|-------------|-------|
| NSP | BERT-base | 110 M | 0.5631 |
| | BERT-large | 340 M | **0.5659** |
| PPL | GPT2 | 117 M | 0.5676 |
| | GPT2-Medium | 345 M | 0.5889 |
| | GPT2-Large | 774 M | 0.5927 |
| | GPT2-XL | 1.6 B | **0.5934** |
| Prompt | GPT3-small | 350 M | 0.5512 |
| | GPT3-Medium | 1.3 B | 0.5318 |
| | GPT3-Large | 6.7 B | 0.5205 |
| | GPT3-Full | 175 B | 0.6358 |
| | GPT4 | >175 B | **0.6574** |

Table 3: Performance with different Model Sizes. NSP, PPL, and Prompt mean the next sentence prediction, perplexity, and prompt model, respectively.

like other models, the AUROC score is equivalent to accuracy. The performances of all models under both settings are presented in Table 2, from which we draw the following observations.

*The corpus statistics method performs poorly.* This could be attributed to two possible reasons: (1) The triggers might not adequately represent the semantics of events, leading to the co-occurrence information between the two triggers being insufficient for predicting the relationship between them; (2) Considering that essentiality knowledge is a form of implicit commonsense knowledge that people seldom discuss, it is difficult to directly identify references to such knowledge within raw corpora.

*Indirect supervision from other NLU tasks proves beneficial.* The experiments involving the TE and QA models demonstrate that fine-tuning with these tasks enhances the language models' ability to better comprehend event essentiality.

*Current NLP models, including the massive GPT models, still fall drastically behind human on ESD.* This implies that the pre-training and fine-tuning over a limited-size dataset might not be enough to uncover the implicit knowledge we need to understand events, which further proves the value of proposing this new and challenging task.

*Almost all models[7] exhibit better performance when the modifier is provided*, which aligns with human performance. This observation suggests that when the description of the goal event is clearer and less ambiguous, the model, similar to humans, can indeed comprehend events more effectively.

## 3.2 Impact of the Model Size and Discussion

To investigate the impact of pre-trained LMs' model sizes on their abilities to understand event

---

[7]The only exception is UnifiedQA.

essentiality, we present the performance of various LM variants in Table 3. The results demonstrate that the model size plays a critical role in the success of these language models. Particularly for GPT-3, reducing the number of parameters to 6.7 billion results in the prompt-based method becoming ineffective. Meanwhile, it raises concerns about the diminishing gain when we further increase the model size. Given that current LMs are already extremely large and expensive to train, it may not be feasible to fully understand events by solely increasing model sizes and corpora. We hope that ESD can promote further research on knowledge acquisition and reasoning, fostering a deeper understanding of events.

## 4 Related Works

The NLP community has increasingly focused on event understanding (Chen et al., 2021), with research divided into event-centric information extraction (IE) (Grishman et al., 2005; Lin et al., 2020; Wang et al., 2020b; Lyu et al., 2021; Zhang et al., 2021b; Feng et al., 2023) and structural event prediction (Zhang et al., 2021a). Event-centric IE includes recognizing, typing events and inferring their relations (Ning et al., 2018b; Glavas et al., 2014; Wang et al., 2023), while structural event prediction involves context-independent inferences about event structures, causality (Gordon et al., 2012; Sap et al., 2019; Li et al., 2021; Zhang et al., 2022), discourse (Chaturvedi et al., 2017), summaries (Li et al., 2021), and memberships (Zhang et al., 2020a; Chen et al., 2020; Zhang et al., 2020b; Wang et al., 2021).

This work pertains to the second research direction, focusing on the internal structure of events rather than relations between them. Unlike previous event membership studies (Zhang et al., 2020a; Chen et al., 2020), this work predicts the essentiality of decomposed subevents, assessing the understanding of internal steps by SOTA LLMs.

## 5 Conclusion

We introduce ESD, an event understanding task assessing state-of-the-art NLP models' comprehension of events by identifying essential ones for goal achievement. Experiments show that complex event knowledge is rarely expressed in text, and current large-scale language models struggle with complex event understanding. We will release all data and code to encourage research on complex

event knowledge collection and improved reasoning for deeper event understanding.

## Acknowledgement

We appreciate the reviewers for their insightful comments and suggestions.

Yueguan Wang was supported by the USC Viterbi-THU Summer Research Program. Muhao Chen is supported by the NSF Grant IIS 2105329, the NSF Grant ITE 2333736, the DARPA MCS program under Contract No. N660011924033 with the United States Office Of Naval Research, a Cisco Research Award, two Amazon Research Awards, and a Keston Research Award.

This research is based upon work supported in part by the Office of the Director of National Intelligence (ODNI), Intelligence Advanced Research Projects Activity (IARPA), via IARPA Contract No. 2022-22072200003 under the HIATUS Program and IARPA Contract No. 2019-19051600006 under the BETTER Program. This work was also supported by Contract FA8750-19-2-1004 with the US Defense Advanced Research Projects Agency (DARPA). Approved for Public Release, Distribution Unlimited. The views and conclusions contained herein are those of the authors and should not be interpreted as necessarily representing the official policies, either expressed or implied, of ODNI, IARPA, DARPA, or the U.S. Government. The U.S. Government is authorized to reproduce and distribute reprints for governmental purposes notwithstanding any copyright annotation therein.

## Ethical Statement

To the best of our knowledge, this work has no ethical concerns. All the collected data are anonymized, and all annotators are paid higher than the minimum payment requirement.

## Limitations

A potential limitation of this work is the data scale, which is not enough for training a decent model. However, as we mainly use the dataset as a test set, the current scale is enough for this purpose.

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

| Model | Encoder Model | # Parameters | Formatted Input | Output Format |
|---|---|---|---|---|
| Next Sentence Prediction | BERT-large | 340 M | **Sentence 1**: grow a magnolia tree; **Sentence 2**: plant the seeds. | Yes/No |
| Perplexity | GPT-2-xl | 1.6 B | **Sentence**: In order to grow a magnolia tree, it is essential to plant the seeds. | Perplexity |
| Intent Detection | RoBERTa-large | 355 M | **prompt**: Plant the seeds **choice**: Grow a magnolia tree | logits score |
| Textual Entailment | RoBERTa-large | 355 M | **Premise**: Grow a magnolia tree **Hypothesis**: Plant the seeds | Entailment/Contradict/Neutral |
| Unified QA | T5-large | 770 M | **Question**: Is it essential to plant the seeds for growing a magnolia tree? | Yes/No |
| Prompt with GPT-3 | GPT-3 | 175 B | **Input**: [Statement]: To grow a Magnolia Tree, you need to plant the seeds. [Answer] | Yes/No |
| Prompt with GPT-4 | GPT-4 | >175 B | **Input**: [Statement]: To grow a Magnolia Tree, you need to plant the seeds. [Answer] | Yes/No |

Table 4: Demonstration of used templates and prompts and summarization of implementation details. The original goal-step pair is ("Grow a magnolia tree", "Plant the seeds "). Templates and prompts used for different models are presented in the "Formatted Input" column.

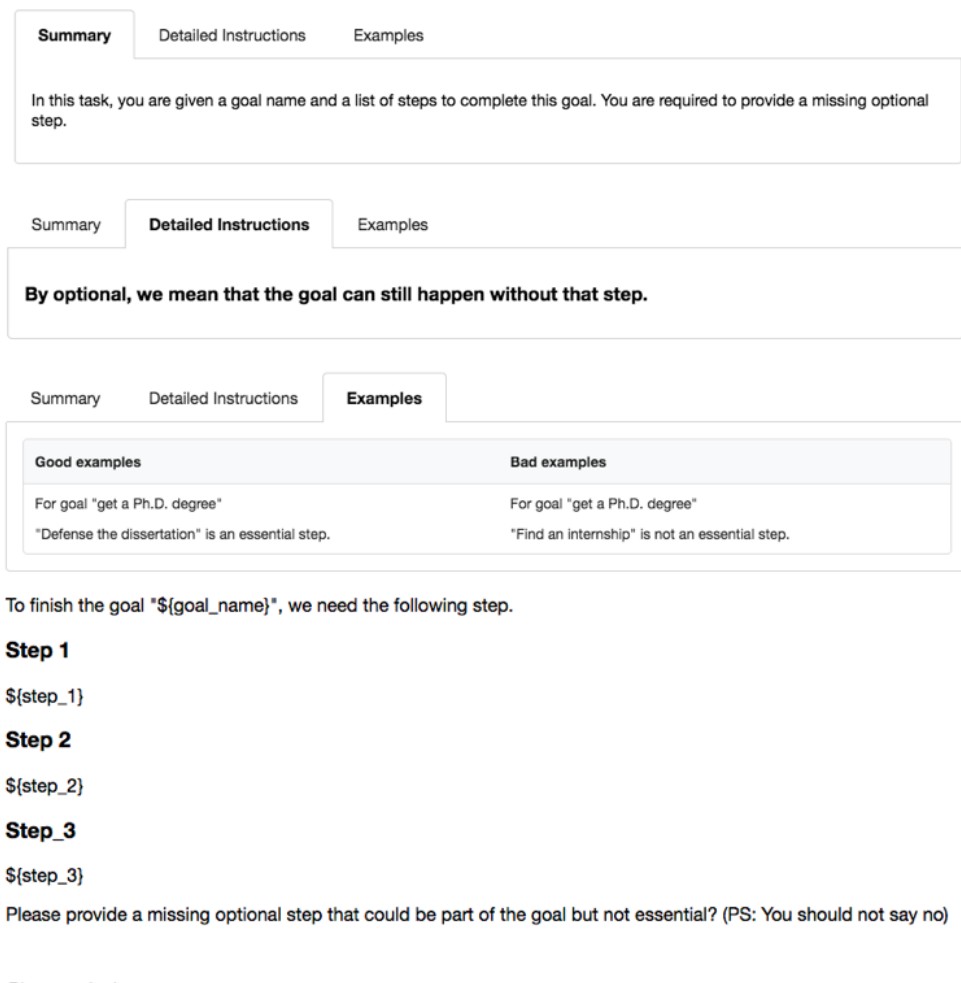

Figure 2: Survey template for adding non-essential steps.

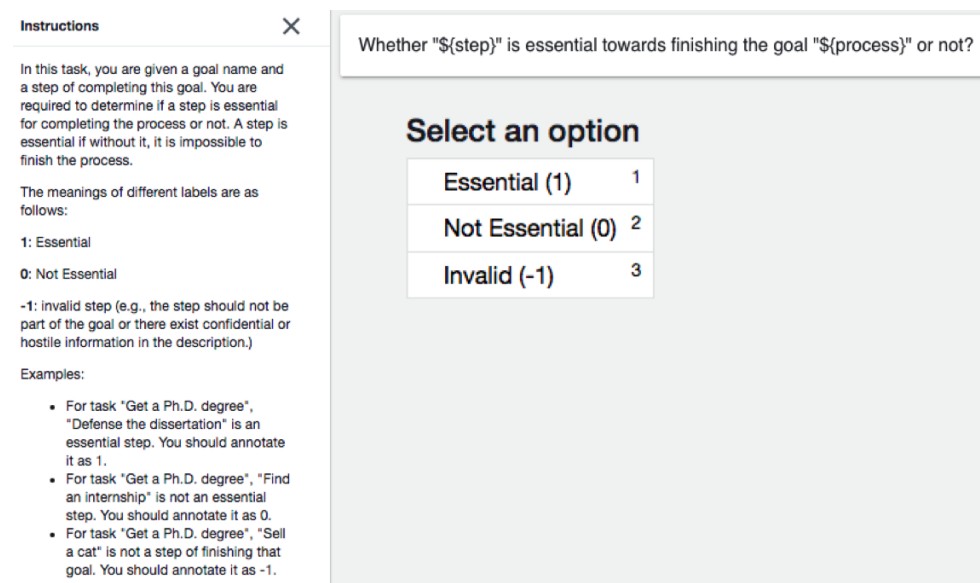

Figure 3: Survey template for annotating the essentiality.