# OpenReview forum: "Are All Steps Equally Important? Benchmarking Essentiality Detection in Event Processes"
_EMNLP/2023/Conference — EMNLP 2023 Main_

### Official Review · Reviewer_piJ4 · 2023-08-02

**Soundness:** 4

**Excitement:**

4: Strong: This paper deepens the understanding of some phenomenon or lowers the barriers to an existing research direction.

**Missing References:**

Previous work on crowdsourcing event steps, which also touches on the optionality of certain steps and the difficulty in obtaining essential steps from corpora
- A Crowdsourced Database of Event Sequence Descriptions for the Acquisition of High-quality Script Knowledge (Wanzare et al., LREC 2016)
- MCScript: A Novel Dataset for Assessing Machine Comprehension Using Script Knowledge (Ostermann et al., LREC 2016)

**Paper Topic And Main Contributions:**

This paper presents work on commonsense events which have a clear goal and in particular at fine-grained steps of these events which are essential to the completion of the goal. They present a new task of ESD, essential step detection, contribute a corpus of goal, step pairs annotated wrt their essentiality relation and evaluate several models to find that this task is indeed very challenging for existing approaches.

**Questions For The Authors:**

- Do you need to differentiate between events which are strongly associated with each other (frequency) and events which have an essentiality relation to each other?
- Was agreement higher on essential or non-essential items? (if you look at high-agreement items, are most of them essential or non-essential?)
- It sounds a bit circular that you use perplexity predicted by a LM as predicted perplexity, can you rephrase this or elaborate on what you mean in terms of human cognition?

**Reasons To Accept:**

- The paper is very clear and the methodology is solid and clearly presented
- The work connects to work in cognitive science about how people deconstruct goal events into smaller steps with a hierarchical structure
-- The topic touches a raw nerve when it comes to limits of LMs and the difference with human performance (which can tap into common sense knowledge)
- Clear human upper bound
- Dataset provided for benchmarking
- Sensible conclusion that essentiality knowledge is implicit / obvious and thus difficult to elicit from corpora

**Reasons To Reject:**

There is some work that already tried to tap into the problem that common sense knowledge does not get explicitly mentioned. I think the authors could refer to that and also (as that work is already rather old) elaborate further and add their own conclusions on what LMs can or cannot achieve using text data when it comes to common sense knowledge.

**Reproducibility:**

5: Could easily reproduce the results.

**Reviewer Confidence:**

5: Positive that my evaluation is correct. I read the paper very carefully and I am very familiar with related work.

---

> ### Author Rebuttal · Authors · 2023-08-29
>
> We would like to thank the reviewer for the comments. We will add the missing references in the final version. Below are the responses to the questions:
>
> Q1. Yes, essentiality and association are different concepts. For example, obtaining a PhD degree in computer science is frequently mentioned with doing an internship during the summer. However, doing an internship is not essential for obtaining a PhD degree.
>
> Q2. The agreement was indeed higher on essential items.
>
> Q3. We used the perplexity score as a proxy for the essentiality of a goal-step pair, assuming that it is more necessary for essential steps to be described in the same context, and therefore have lower perplexity scores.

---

### Official Review · Reviewer_eKZY · 2023-08-04

**Soundness:** 4

**Excitement:**

4: Strong: This paper deepens the understanding of some phenomenon or lowers the barriers to an existing research direction.

**Paper Topic And Main Contributions:**

This paper introduces an enhanced dataset based on WikiHow, where the steps are manually annotated with their essentiality. It creates a novel task of identifying which steps are necessary in an event process. The study explores various methods to accomplish this task and compares their results.

**Reasons To Accept:**

1. The paper introduces an enhanced dataset based on WikiHow and proposes a new task of recognizing essential steps.
2. Various methods were attempted to accomplish this task, and the results were compared, confirming significant differences between the model performance and human performance on this task.

**Reasons To Reject:**

Additional experiments could be considered, using open-source large-scale models like LLAMA to further explore the task.
A more detailed result analysis could be conducted specifically for the results obtained with the LLAMA model.





**Reproducibility:**

4: Could mostly reproduce the results, but there may be some variation because of sample variance or minor variations in their interpretation of the protocol or method.

**Reviewer Confidence:**

4: Quite sure. I tried to check the important points carefully. It's unlikely, though conceivable, that I missed something that should affect my ratings.

---

> ### Author Rebuttal · Authors · 2023-08-29
>
> We would like to thank the reviewer for the comments. We will add results and discussions with the LLaMa model in the final version.

---

### Official Review · Reviewer_PfMd · 2023-08-07

**Soundness:** 3

**Excitement:**

3: Ambivalent: It has merits (e.g., it reports state-of-the-art results, the idea is nice), but there are key weaknesses (e.g., it describes incremental work), and it can significantly benefit from another round of revision. However, I won't object to accepting it if my co-reviewers champion it.

**Paper Topic And Main Contributions:**

This paper introduces a new benchmark dataset, Essential Step Detection (ESD). It utilizes the WikiHow dataset and involves three annotators labeling whether the sub-step of goal G is essential or not. To evaluate the dataset's difficulty, the authors conducted experiments using various models and tasks. The results indicate that current models are incapable of differentiating essential steps corresponding to goals, suggesting a new research problem for language models.

**Questions For The Authors:**

* Is there a threshold for determining a low PPL (Perplexity) in the evaluation? The other models and tasks are classification tasks, where predictions can be discretely labeled.
* When forming the dataset, is the order of the steps considered? (i.e., priority of steps that must be done first)
* Regarding the 1,000 random goal-oriented processes from WikiHow, does this number represent the total number of goals or the approximate total number of steps? If it is the number of goals and each goal consists of three to ten steps, shouldn't there be more than 1,500 instances?
* Could you provide examples of instances that are selected as not essential or invalid?

**Reasons To Accept:**

Recent works, such as ChatGPT, have incorporated human preference into language models, resulting in an enhancement of the generated output's quality. Besides explicit training for natural language understanding and generation, capturing implicit facts or meanings that are evident to humans remains a challenge. To design robust and human-like language models, I agree with the authors on the necessity of creating new evaluation datasets to measure their abilities.

**Reasons To Reject:**

This paper introduces an intriguing dataset and task that language models should be capable of performing as well as humans. However, it does not provide a simple solution to this problem or demonstrate how discriminating essential steps can improve the performance of language models. For example, it does not show empirical evidence that using only essential steps can enhance WikiHow QA performance.

**Reproducibility:**

4: Could mostly reproduce the results, but there may be some variation because of sample variance or minor variations in their interpretation of the protocol or method.

**Reviewer Confidence:**

3: Pretty sure, but there's a chance I missed something. Although I have a good feel for this area in general, I did not carefully check the paper's details, e.g., the math, experimental design, or novelty.

---

> ### Author Rebuttal · Authors · 2023-08-29
>
> We appreciate the reviewer’s comments. Below are the responses to the questions:
>
> RR. How discriminating essential steps can improve the performance of language models: the experiments involving the TE and QA models demonstrate that fine-tuning with these tasks enhances the language models’ ability to better comprehend event essentiality (line 256).
>
> Q1. Threshold for determining a low PPL: we evaluate all models based on the AUROC score, so we did not set a specific threshold for perplexity.
>
> Q2. Order of the steps are not considered during the annotation. The annotators only see a pair of goal and sub-step and are asked whether this sub-step is essential to it.
>
> Q3. For those initial 1,000 goals selected, we only kept those with three to ten steps to avoid too simple/complicated processes, resulting in about 400 different goals in the dataset.
>
> Q4. Here are some examples that are selected as non-essential:
> (1) Goal: Open CSV Files - OpenOffice Calc; Sub-step: Select the “Separate By” radio button
> (2) Goal: Sew on a Wig - Preparing Your Hair and the Wig; Sub-step: Braid your hair
> (3) Goal: Draw Cute Animals - A Cute Tiger; Sub-step: Color your drawing

---

### Meta-Review · Area_Chair_prQz · 2023-09-19

**Recommendation:** 4

**Metareview:**

The paper presents an event essentiality task with respect to a goal. It introduces a dataset and benchmarks the performance of various existing models on this task.

The paper is has many positives. First, detecting essential events is well-motivated. Not all events in steps that lead to a goal are essential. The task stands on its own for event and goal related reasoning but can be useful for downstream applications as well. The dataset is a useful addition to the broad collection of event related datasets. The paper is written clearly (given the page limits of a short paper). The primary significance of this work is the potential for improving and evaluating event essentiality reasoning in language models using this resource.

The main weaknesses are that the downstream uses of the dataset or the task is unclear. This is indeed a challenge for many event related resources and not just this specific resource. In addition to the presentation and clarity questions raised by the reviewers, I have one additional concern. The definition of what makes something essential, while it sounds reasonable, can be hard to pin down in specific cases. There is not much discussion in the paper about this aspect.

---

### Decision · Program_Chairs · 2023-10-07

**Decision:**

Accept-Main

**Comment:**

The paper presents an event essentiality task with respect to a goal. It introduces a dataset and benchmarks the performance of various existing models on this task.

The paper is has many positives. First, detecting essential events is well-motivated. Not all events in steps that lead to a goal are essential. The task stands on its own for event and goal related reasoning but can be useful for downstream applications as well. The dataset is a useful addition to the broad collection of event related datasets. The paper is written clearly (given the page limits of a short paper). The primary significance of this work is the potential for improving and evaluating event essentiality reasoning in language models using this resource.

The main weaknesses are that the downstream uses of the dataset or the task is unclear. This is indeed a challenge for many event related resources and not just this specific resource. In addition to the presentation and clarity questions raised by the reviewers, I have one additional concern. The definition of what makes something essential, while it sounds reasonable, can be hard to pin down in specific cases. There is not much discussion in the paper about this aspect.